# The Current State of Mock Circulatory Loop Applications in Aortic and Cardiovascular Research: A Scoping Review

**DOI:** 10.3390/biomedicines14010028

**Published:** 2025-12-22

**Authors:** Felix E. N. Osinga, Nesar A. Hasami, Jasper F. de Kort, Emma-Lena Maris, Maurizio Domanin, Martina Schembri, Alessandro Caimi, Michele Conti, Constantijn E. V. B. Hazenberg, Ferdinando Auricchio, Jorg L. de Bruin, Joost A. van Herwaarden, Santi Trimarchi

**Affiliations:** 1Section of Vascular Surgery, Cardio Thoracic Vascular Department, Fondazione IRCCS Ca’ Granda Ospedale Maggiore Policlinico, 20122 Milan, Italy; 2Department of Vascular Surgery, Erasmus University Medical Centre, 3015 GD Rotterdam, The Netherlands; 3Department of Cardiothoracic Surgery, Radboud University Medical Centre, 6525 GA Nijmegen, The Netherlands; 4Department of Vascular Surgery, University Medical Centre Utrecht, 3584 CX Utrecht, The Netherlands; 5Department of Clinical Sciences and Community Health, Università degli Studi di Milano, 20122 Milan, Italy; 6Civil Engineering and Architecture Department, Università degli Studi di Pavia, 27100 Pavia, Italy; 73D and Computer Simulation Laboratory, IRCCS Policlinico San Donato, 20097 Milan, Italy

**Keywords:** mock circulatory loop, aorta, endovascular aortic repair, TEVAR, pulse wave velocity, aortic stiffness

## Abstract

**Background**: Mock circulatory loops (MCLs) are benchtop experimental platforms that reproduce key features of the human cardiovascular system, providing a safe, controlled, and reproducible environment for haemodynamic investigation. This scoping review aims to systematically map the current landscape of MCLs used for aortic simulation and identify major areas of application. **Methods**: A systematic search of PubMed, Scopus, and Web of Science identified original studies employing MCLs for aortic simulation. Eligible studies were categorized into predefined themes: (I) (bio)mechanical aortic characterization, (II) hemodynamics, (III) device testing, (IV) diagnostics, and (V) training. Data on MCL configurations, aortic models, and study objectives were synthesized narratively. **Results**: Eighty-four studies met the inclusion criteria. Twenty-five investigated aortic biomechanics, 23 hemodynamics, 22 device or product testing, 13 validated diagnostic imaging techniques, and one training application. Models included porcine (*n* = 22), human cadaveric (*n* = 7), canine (*n* = 1), ovine (*n* = 1), bovine (*n* = 1), and 3D-printed or molded aortic phantoms (*n* = 55). MCLs were employed to study parameters such as aortic stiffness, flow dynamics, dissection propagation, endoleaks, imaging accuracy, and device performance. **Conclusions**: This review provides a comprehensive overview of MCL applications in aortic research. MCLs represent a versatile pre-clinical platform for studying aortic pathophysiology and testing endovascular therapies under controlled conditions. Standardized reporting frameworks are now required to improve reproducibility and accelerate translation to patient-specific planning.

## 1. Introduction

Mock Circulatory Loops (MCLs) are experimental benchtop models designed to simulate the human cardiovascular system (CVS) under both physiological and pathological conditions (see Figure 1). MCLs can mimic hemodynamic parameters such as heart rate, cardiac output, cardiac contractility, arterial compliance, peripheral resistance, and fluid inertance in vitro or ex vivo. In contrast to in vivo studies, in vitro and ex vivo experiments are preferred when precise quantitative analysis and control of physiological parameters are required, for pre-clinical device testing, and to minimize the use of animal studies.

MCLs have evolved from simple hydraulic testbeds to sophisticated hybrid platforms integrating real-time control systems and anatomically accurate components. This development enables simultaneous, accurate, and repeatable measurements of (i) hemodynamics, including flow rate and pressure, and (ii) aortic wall characteristics, such as stiffness, displacement, and deformation. Moreover, they play a key role in the design, development, and in vitro assessment of implantable aortic devices (such as stent-grafts and intra-aortic balloon pumps), surgical treatments, and diagnostic imaging techniques.

Despite these advancements, no comprehensive synthesis currently exists regarding the application of MCLs in aortic research. This scoping review aims to systematically map the current state of MCL-based aortic simulation and to identify unmet methodological and reporting gaps relevant to translational and clinical implementation.

## 2. Materials and Methods

### 2.1. Review Design

This scoping review followed the Preferred Reporting Items for Systematic Reviews and Meta-Analyses Extension for Scoping Reviews (PRISMA-ScR) and was conducted according to established methodological frameworks [1,2]. The protocol was registered and made publicly available on the Open Science Framework: 10.17605/OSF.IO/Z37FY.

### 2.2. Objectives

The objective of this scoping review was to systematically map and synthesize the methods, applications, and outcomes of studies employing MCLs for aortic simulation and identify gaps in knowledge.

### 2.3. Literature Sources and Search Strategy

Two authors (F.E.N.O. and J.F.d.K.) independently conducted the literature search and study selection. In cases of discrepancies, a third and/or fourth senior author (J.A.v.H. and S.T.) was consulted to provide consensus.

A comprehensive search of PubMed, Scopus, and Web of Science was performed on 7 July 2025 (Appendix A). The search string was designed iteratively by the review team (F.E.N.O. and J.F.d.K.) in consultation with senior authors (J.A.v.H. and S.T.) and was structured around two core categories, (i) MCLs and (ii) the aorta, with additional terms reflecting device testing, biomechanical properties, and model types. Additionally, backward and forward citation tracking was performed.

### 2.4. Inclusion and Exclusion Criteria

All original English-language articles presenting a simulation model of the aorta, or parts of it, integrated into an MCL were included. Review articles, commentaries, abstracts without full text, and purely computational (in silico) studies were excluded.

### 2.5. Study Selection

All search results were imported into Rayyan.ai for organization and screening [3]. No automation or AI-assisted tools were used during the selection process. After removal of duplicates, titles and abstracts were screened manually. Full-text articles were retrieved for all potentially relevant studies. Reference lists of included studies were reviewed for additional eligible publications. The study selection phase was concluded on 15 September 2025.

### 2.6. Data Acquisition and Categorization

Data extraction was performed independently by two reviewers (F.E.N.O. and N.A.H.) using a predefined extraction form in Microsoft Excel (Microsoft Corp., Redmond, WA, USA). To facilitate structured comparison across the included studies, the following themes were established. Each study was discussed by three authors (F.E.N.O.; N.A.H.; and J.F.d.K.), and subsequently classified into predefined themes: (I) aortic characteristics, referring to physiological or post-intervention (bio)mechanics of the aortic wall; (II) aortic hemodynamics, focusing on flow behavior and pressure dynamics; (III) device testing, encompassing surgical, radiological, and phantom-based evaluations of products; (IV) diagnostics, addressing image-based methods for assessment and quantification of aortic pathology; and (V) training, involving the use of MCLs for procedural simulation and skill development.

### 2.7. Core Components of Mock Circulatory Loops

Systemic circulation MCLs typically consist of two primary elements: a flow driver and a closed circulatory loop. The driver generates continuous or pulsatile flow. In pulsatile systems, stroke volume and heart rate can be adjusted to simulate cardiac function. Various pump types are used in MCLs. Piston and diaphragm (sac-type) pumps generate controlled pulsatile flow that mimics ventricular contractions. Roller (peristaltic) pumps are widely used in current cardiac support systems and provide easy-to-control and reliable flow through flexible tubing with intrinsic backflow protection. Centrifugal pumps deliver continuous, smooth flow for non-pulsatile simulations. Gear pumps enable precise volumetric control under high pressures. Lastly, indirect compression systems reproduce ventricular motion by compressing compliant chambers using pneumatic, hydraulic, or servo-driven actuators [4].

To mimic preload, a venous compliance chamber or reservoir can be positioned upstream of the pump to represent the venous system. This reservoir collects the working fluid at the end of the loop and recirculates it, ensuring volume stability and continuous closed-loop operation. Ideally, the reservoir can be adjusted via pressure or fluid height to regulate ventricular filling conditions, i.e., preload.

To reproduce afterload, the system can incorporate an arterial compliance chamber with or without a peripheral resistance element. The arterial compliance chamber represents the elastic properties of the arterial walls, absorbing the pulsatile output of the pump and generating a physiologic pressure-flow relationship (the Wind Kessel effect). By adjusting the arterial compliance chamber, different levels of arterial stiffness can be achieved. A peripheral resistance element simulates systemic vascular resistance and can be adjusted to mimic varying levels of vasoconstriction and vasodilation to represent specific vascular beds within the circulation.

### 2.8. Data Presentation

The extracted data were summarized in textual and tabular formats. Data were presented as reported in the original studies, without additional statistical transformation. Categorical variables were presented as absolute numbers (*n*) and percentages (%), and continuous variables were reported as mean ± standard deviation (SD). Where available, ranges were also provided. Missing or unavailable data were denoted as (–). No statistical pooling or meta-analysis was performed due to heterogeneity among studies.

## 3. Results

### 3.1. Study Selection

The initial search yielded 2063 studies across three databases. Duplicate removal resulted in 1093 studies to screen on title and abstract. Eighty-one full-text articles were retrieved, of which four could not be obtained. A total of 77 full-text articles were assessed, and 69 met the inclusion criteria. An additional 15 studies were identified through backward and forward citation, resulting in a final total of 84 included studies. Details and study characteristics are provided in Appendix A, and Figure 2 provides the detailed flow diagram of study selection.

### 3.2. Study Characteristics and Publication Trends

Most included articles originated from Europe and North America, with the largest contributions from the USA (*n* = 20; 23.8%) [5,6,7,8,9,10,11,12,13,14,15,16,17,18,19,20,21,22,23,24], followed by Italy (*n* = 9; 10.7%) [25,26,27,28,29,30,31,32,33], The Netherlands (*n* = 9; 10.7%) [34,35,36,37,38,39,40,41,42], Austria (*n* = 6; 7.1%) [43,44,45,46,47,48], Canada (*n* = 6; 7.1%) [49,50,51,52,53,54], France (*n* = 6; 7.1%) [55,56,57,58,59,60], and Germany (*n* = 6; 7.1%) [61,62,63,64,65,66].

The first aortic MCL simulation study was published in 1998 in *The Journal of Vascular Surgery* by Schurink et al. [41]. Studies published between 1998 and 2025 were included with various themes, summarized in Table 1. Notably, 36 of 84 (42.9%) studies were published in the last 5 years. Mann–Kendall test confirmed a statistically significant upward publication trend from 2016 (τ = 0.488, *p* = 0.003), as shown in Figure 3.

Of the 84 included studies, 48 (57.1%) were published in clinically oriented journals, while 30 (35.7%) were published in engineering journals. Only three studies (3.6%) each were published in broad-scope or physics journals.

### 3.3. Characteristics of Mock Circulatory Loops

The main characteristics of the mock circulatory loops used in the included studies are summarized in Table 2.

#### 3.3.1. Flow Patterns and Pump Types

Most of the studies employed pulsatile flow (*n* = 75; 90.5%); six studies [6,15,27,29,39,56] used a continuous flow; two studies [8,52] used a pulsatile and (quasi)static flow. One study did not specify the flow pattern [45]. Several types of pumps were employed to generate flow within the MCLs: piston pumps (*n* = 43; 51.2%) were most common, followed by diaphragm or sac-type pumps (*n* = 8; 9.5%), gear pumps (*n* = 6; 7.1%), indirect compression systems (*n* = 5; 6.0%), centrifugal pumps (*n* = 4; 4.8%), and roller (peristaltic) pumps (*n* = 3; 3.6%). One study used a continuous-flow pump with a magnetic valve to create pulsatile flow. In 14 studies, the pump type was not reported.

#### 3.3.2. Type of Aortic Models

Three studies employed hybrid experimental set-ups combining biological and synthetic; two combined phantom and porcine aortas [6,67], the other combined phantom and bovine aortas [18].

Twenty-nine (34.5%) used exclusively biological tissue, including seven with human cadaveric aortas [24,43,47,50,52,55,57], one with canine [16], and one with ovine and porcine [46]. The remaining 20 biological studies used porcine aortas [8,11,15,22,26,27,28,29,30,31,32,33,34,35,36,42,44,48,65,68].

In contrast, 52 studies employed synthetic aortic models, 20 of which were patient-specific phantoms fabricated via 3D printing or mold-based casting [5,12,14,17,21,23,25,37,45,56,59,60,62,63,66,69,70,71,72,73].

#### 3.3.3. Type and Temperature of Working Fluid

In 33 studies (39.3%), glycerol or glycerin mixed with water or saline was used as the perfusion fluid. Fourteen studies (16.7%) employed saline, sometimes phosphate-buffered, aerated with 5% CO_2_/95% O_2_, or supplemented with contrast agents. Twenty-two studies (26.2%) utilized water as the perfusate. Four studies applied a commercially available blood-mimicking fluid [9,11,22,71]. Schurink et al. [41] used a starch solution with plasma-like viscosity, while Qing et al. [68] employed Gelofusine. Only Desai et al. [67] used anticoagulated blood for the perfusion of porcine aortas, and for synthetic aortas, they prepared a complex medium consisting of 8% low-molecular-weight Dextran (77 kDa) dissolved in M199 minimum essential medium, supplemented with 20% fetal bovine serum (FBS), 7.5% sodium bicarbonate, 200 mmol l-glutamine, and the antibiotics penicillin (10,000 U/mL) and streptomycin (10 mg/mL), adjusted to pH 7.20 ± 0.01. Eight studies did not report the working fluid [14,44,45,54,58,60,74,75].

A total of 37 studies reported the temperature of the working fluid, which was 37 °C in 27 studies [7,9,18,24,26,27,28,29,30,31,34,40,42,43,45,46,47,49,50,51,52,65,67,68,71,72,73].

#### 3.3.4. Thoracic and Abdominal Environment

Thirteen articles reported a type of thoracic and/or abdominal environment simulation. Four studies mounted the aortic phantoms on a 3D-printed spine to replicate the mechanical support provided by the surrounding anatomy. Three of these used gelatine solutions [35,36,42], while one applied water [60] to mimic abdominal pressure. One study embedded a vascular model in a radiopaque anthropomorphic body phantom with a skeleton [66]. Other studies immersed the aortas in saline [67,76], glycerin [43], a glycerol-water solution [47], or a tissue-mimicking gel [6,8]. One study positioned an aortic endoleak model within a customized abdominal phantom that included tissue-mimicking material and representations of the spine, kidneys, liver, and spleen to create a realistic imaging background during fluoroscopy and CT [40]. Another study embedded the aortic phantom in a gel block to provide a static tissue reference [17].

#### 3.3.5. Study Themes

##### 3.3.5.1. Aortic (Bio)mechanical Characteristics

A total of 25 studies investigated the (bio)mechanical characteristics of the aorta, including 12 non-interventional and 13 post-interventional models.

Non-Interventional Characterization

Several studies investigated the intrinsic mechanical behavior of the native or untreated aorta, mainly viscoelastic and mechanical properties. Amabili et al. [49] and Franchini et al. [52] quantified the viscoelastic dynamics of the human descending thoracic aorta, reporting dynamic stiffness ratios and energy loss factors. Mascarenhas et al. [34] combined 2D ultrasound elastography with inflation testing to estimate the mechanical properties of porcine aortas.

Ene et al. [77] evaluated the mechanical impact of intraluminal thrombus in abdominal aortic aneurysm (AAA) phantoms by testing silicone models with varying intraluminal thrombus stiffness and thickness, showing that the presence and thickness of intraluminal thrombus reduced compliance and shifted maximal wall strain proximally. Wen et al. [73] combined phase-contrast magnetic resonance imaging (MRI) and computational fluid dynamics to determine flow characteristics and wall dynamics, finding that regions of high wall shear stress and pressure align with dissection-prone sites, whereas low wall shear stress and high oscillatory shear index are linked to wall thickening. Cameron et al. [53] assessed the effect of aortic compliance on pump performance, confirming that compliant mock aortas generated more physiologic waveforms, steadier flow, and reduced cardiac workload compared to rigid models.

In addition, several studies focused specifically on aortic stiffness. Mandigers et al. [31] investigated the association between aortic arch geometry and aortic stiffness, demonstrating that increased arch angulation is associated with higher pulse wave velocity (PWV), blood pressures, and amplifies the stiffening effect of stent-grafts. Jarman et al. [46] investigated the effect of different loading conditions on aortic stiffness indices and concluded that aortic elasticity is highly dependent on blood pressure. Complementary imaging-based approaches were reported by Boese et al. [61], Gaddum et al. [74], and Moravia et al. [59]. Boese et al. [61] evaluated the feasibility and accuracy of measuring PWV using phase-contrast MRI techniques. Gaddum et al. [74] validated a real-time MRI sequence to quantify beat-to-beat variations in PWV during respiratory maneuvers. Moravia et al. [59] applied particle image velocimetry (PIV) to assess PWV and corresponding stiffness using the lnD–U method, a one-point technique that relates instantaneous changes in flow velocity to logarithmic variations in aortic diameter, enabling localized stiffness assessment without dual-point pressure measurements. Zimmermann et al. [5] further examined the impact of vessel wall stiffness on 4D-flow MRI hemodynamic quantification.

Post-Interventional Characterization

Several studies investigated the biomechanical effects of endovascular interventions in human and porcine aortas. Agrafiotis et al. [43] investigated the impact of thoracic endovascular aortic repair (TEVAR) in human thoracic aortas, showing that stent-graft implantation significantly reduced distensibility and increased stiffness. Nauta et al. [27,29] assessed longitudinal [29] and radial [27] strain after TEVAR in porcine aortas, showing significant reductions in both longitudinal and radial strain. Similarly, in porcine models, de Beaufort et al. [26,28] and Bianchi et al. [33] demonstrated increased PWV following stent-graft deployment for TEVAR, with effect depending on the extent of stent graft coverage, while Mandigers et al. [30] confirmed the stiffening effects when comparing two generations of Valiant grafts. Schellinger et al. [65] determined that endograft placement in the abdominal aorta induced stiffness gradients and altered circumferential strain.

Yusefi et al. [47] bridged the comparison with open arch replacement (OAR) by evaluating both OAR and TEVAR in human aortas, showing that TEVAR led to higher systolic pressure and input impedance, indicating increased left ventricular afterload and altered wave dynamics. De Kort et al. [32] extended this to open surgical repair, showing increased PWV after Dacron graft replacement in a porcine descending aorta.

Other in vitro experiments focused on device design. Morris et al. [72] compared four commercial abdominal stent-grafts with a multilayer flow modulator, showing that most grafts reduced compliance and increased stiffness, while the multilayer device preserved compliance best. Moreover, Legerer et al. [78] tested an external spring casing around Dacron grafts, aiming to restore aortic compliance after open surgical repair, which reduced PWV and pulse pressure.

Lastly, Timaran et al. [10] showed that increasing type II endoleak volume raised aneurysm wall pressure, especially at the site of maximum diameter.

##### 3.3.5.2. Hemodynamics

A total of 23 studies investigated the aortic hemodynamics within MCLs, focusing on aortic dissection dynamics, flow behavior, pressure distribution, and treatment-related hemodynamic changes. Ten studies did an intervention, while thirteen studies did not perform an intervention; two of those were in healthy aortas and eleven in pathologic aortas.

Several of these studies explored the propagation mechanisms of type B aortic dissections (TBAD), demonstrating how primary entry tear size [9], configuration [79], location [44,57], pulse pressure [11], and re-entry tears [22] influence false lumen pressurization and dissection flap dynamics. Birjniuk et al. [23] used 4-dimensional phase-contrast MRI and flexible aortic models to reveal significant flow reversal, vortices, and flap motion within the false lumen. Chung et al. [19,20] demonstrated that true-lumen collapse in TBAD strongly depends on the difference in ratios of inflow capacity to outflow capacity in the true and false lumina, both anatomic and physiologic factors can affect true-lumen collapse [19], and that covering the primary entry tear with a stent-graft is the most effective way to relieve true-lumen collapse [20]. Morris et al. [71] developed a patient-specific in vitro model of TBAD to assess the hemodynamics within the true and false lumina.

Nair et al. [12] combined an MCL with fluid–structure interaction simulations to evaluate how exercise affects pressure gradients across aortic coarctations, showing that pressure drop increases nonlinearly with cardiac output.

Five studies investigated the effects of certain interventions on aortic hemodynamics. Ong et al. [80] assessed a novel stent graft design with slit perforations in thoracic aortic aneurysm repair that can positively alter the hemodynamics at the aortic arch while maintaining blood flow to supra-aortic branches. Morris et al. [70] investigated the effect of variations in stent-graft types, demonstrating that commercial stent-grafts reduce aortic compliance, increase PWV, cause pressure wave reflections, and disturb flow at the bifurcation, while a tapered geometry improved flow uniformity. Pasta et al. [25] investigated how excessive bird-beak configurations in thoracic endografts influence the risk of infolding using a patient-specific pulsatile flow phantom. Williamson et al. [81] developed a silicone aortic phantom with a surrogate Frozen Elephant Trunk to study Type 1B endoleaks using PIV. Qing et al. [68] demonstrated that after TEVAR for TBAD in porcine aortas, the false lumen can remain pressurized despite successful closure of the primary entry tear. Moreover, nearly 80% of the pressure remained in the thrombosed false lumen. Lastly, Mirgolbabee et al. [37] used contrast-enhanced ultrasound PIV in a patient-specific phantom to study limb thrombosis after endovascular aortic repair.

Schurink et al. [41] and Blackwood et al. [21] also investigated endoleaks and their relation to aortic hemodynamics. Schurink et al. [41] found in an in vitro aneurysm model that every endoleak caused pressure greater than the systemic diastolic pressure within the aneurysmal sac, and all endoleaks could be visualized with a delayed computed tomography angiography (CTA) protocol. Blackwood et al. [21] demonstrated that endotension can result from a non-visualized type Ia endoleak with no flow, while also confirming the findings of Schurink et al.

From a methodological perspective, three studies applied PIV. Yazdi et al. [82] investigated the effect of compliance on aortic arch hemodynamics experiencing pulsatility, finding that compliant aortic phantoms exhibit larger and more irregular recirculation zones than rigid ones, especially under deceleration, in regions earlier identified as high-risk areas for atherosclerosis. Büsen et al. [62] studied the effects of aortic stiffness on hemodynamics in a compliant silicone aortic phantom, demonstrating that reduced compliance increased pressure change rates and mean flow velocity while diminishing vortex formation. Steinlauf et al. [83] examined the effects of the treatment approach on unsteady hemodynamics and blood perfusion to the upper vessels in models of an aortic arch aneurysm, and of the three common repair approaches: open-chest surgical repair, chimney, and hybrid approach.

Lastly, Chen et al. [69] developed a patient-specific MCL to simulate normal and dissected aortic flow in silicone aortic phantoms, which reproduced physiological pressures and flow distributions, revealing higher true lumen velocities and pressure differences in aortic dissection models.

##### 3.3.5.3. Device Testing

Twenty-two studies were conducted to evaluate devices or products, including 11 surgical devices, 6 radiological devices, and 5 phantoms.

Surgical Device Testing

Several studies focused on surgical devices for aortic interventions. Stent-graft fixation and oversizing were both investigated by Kratzberg et al. [18] and Canaud et al. [55] Kratzberg et al. examined the role of barbs and oversizing of abdominal grafts in bovine aortas and synthetic phantoms [18], finding that excessive oversizing compromised fixation and increased risk of migration. Canaud et al. investigated the impact of aortic arch angulation on thoracic stent-graft apposition in human cadaveric aortas [55], showing that radial force and open stent segments were more important for secure fixation than hooks alone.

Mechanical properties of graft materials were assessed by Amabili et al. [49] and Ferrari et al. [51], who characterized the viscoelastic and dynamic behavior of Dacron grafts. Hauck et al. [45] and Zimpfer et al. [48] assessed the feasibility of novel aortic endografts in phantom and porcine ascending aortas, respectively.

Three studies focused on improving the safety and performance of resuscitative endovascular balloon occlusion of the aorta. Maleckis et al. [24] found that safe balloon inflation parameters in cadaveric human aortas depend on anatomical, physiological, and demographic characteristics, and that pressure-guided rather than volume-guided inflation may reduce aortic rupture risk. Madurska et al. [15] compared inflation characteristics and rupture risk of compliant versus semi-compliant balloons in porcine aortas. Lastly, McCarthy et al. [58] introduced an intelligent automated balloon pressure management system, showing that the technique is feasible in porcine aortas.

Jansen et al. [39] preclinically evaluated Fiber Optic RealShape (FORS) technology, which enables real-time 3-dimensional visualization of endovascular devices without fluoroscopy, demonstrating the safety and feasibility in phantom models.

Finally, a novel injectable hydrogel to improve endoleak embolization after EVAR was evaluated by Zehtabi et al. [7], showing successful occlusion in an aortic phantom model.

Radiological Device Testing

Six studies focused on radiological devices and image-guidance technologies. Nagel et al. [40] optimized the radiopacity of a novel injectable polymer for type II endoleak treatment, aiming to identify the minimum tantalum concentration sufficient for safe injection into the aneurysmal sac. Two studies by Riga et al. [84,85] tested remotely steerable robotic catheters in arch cannulation [84] and fenestrated stent-grafting [85], showing shorter cannulation times, fewer movements, and reduced radiation exposure compared with conventional techniques. Sidhu et al. [86] evaluated an electromagnetic 3-dimensional navigation system for arterial cannulation, demonstrating feasibility and a significant reduction in fluoroscopy use during arterial cannulation tasks. Rengier et al. [64] assessed the effect of nitinol stent grafts on flow measurements by 3-dimensional velocity-encoded cine MRI, revealing a systematic positive bias but overall feasibility of in-stent flow quantification. Lan et al. [17] validated a reduced unified continuum formulation for fluid–structure interaction against in vitro 4-dimensional flow MRI in a patient-specific compliant phantom.

Phantom Validation for Radiology and Training

Five studies focused on physiologically realistic aortic phantoms for device evaluation, imaging, and surgical purposes. Mirgolbabee et al. [38] introduced a framework to generate a 50-patient cohort-averaged AAA phantom, showing a good match with its reference model by fabricating a 3D-printed thin-walled phantom and demonstrating the feasibility of flow-field quantification within the phantom. Desai et al. [67] manufactured a physiologically relevant nanocomposite aortic model designed for long-term fatigue analysis of stent-grafts, outperforming conventional latex models. Urbina et al. [87] developed an MRI-compatible phantom to simulate normal and coarctation conditions, with flow and pressure measurements aligning closely with healthy volunteers and patient data. Perrot et al. [60] developed a patient-specific abdominal aortic aneurysm phantom to reproduce realistic biomechanical behavior. Through combining ultrasound and digital image stereo-correlation, they validated the setup as a reliable platform to study vessel wall motion and potential complications of endovascular aortic repair in a controlled environment. Finally, Mohl et al. [63] created a perfused, patient-specific TBAD phantom with a flexible dissection flap, enabling the simulation of TEVAR procedures in a real endovascular operating environment.

##### 3.3.5.4. Diagnostics—Image Validation

Thirteen studies had the aim of testing diagnostics or image validation. Li et al. [16] aimed to improve the clinical utility of Pulse Wave Imaging using conventional ultrasound (US) by evaluating spatial or temporal resolution in ex vivo canine aortas. Building upon these efforts in ultrasound-based assessment, de Hoop et al. demonstrated that multi-perspective acquisitions with dual transducers improved motion tracking and strain estimation in porcine aortas [36] and later expanded this concept to bistatic 3-dimensional acquisitions, further enhancing motion tracking accuracy and spatial resolution for vascular elastography [35]. Van Disseldorp et al. [42] validated a 4-dimensional US-based non-invasive aortic stiffness quantification in porcine aortas, which was highly reproducible and showed moderate agreement with biaxial tensile testing of excised aortic tissue. Wang et al. [54] validated an ultrafast principal strain estimator combining tissue-Doppler Imaging and optical flow tracking to evaluate aortic stiffness, demonstrating that the method can detect and evaluate aortic aneurysm stiffness.

MRI–based methods were also validated in MCLs. Roberts et al. [75] proved the feasibility of real-time cardiovascular MRI to measure PWV during exercise stress testing by validating it against standard phase-contrast MRI in phantoms, while Xu et al. [8] and Zhang et al. [6] validated magnetic resonance elastography to assess global and regional aortic stiffness, respectively, in phantoms and porcine aortas. Potthast et al. [88] validated intra-arterial magnetic resonance aortography with high-speed parallel acquisition in phantoms, reducing contrast dose and scan time without loss of contrast-to-noise ratio.

Other groups focused on imaging accuracy for interventions. Moresco et al. [13] compared CO_2_ angiography with iodinated contrast and intravascular ultrasound in a flow phantom, demonstrating that CO_2_ angiography consistently yields significantly larger vessel measurements. Sieren et al. [66] tested fusion imaging registration in a pulsatile anthropomorphic aortic phantom and showed that 3D–3D methods were significantly more accurate than 2D–3D methods, with computational tomography and magnetic resonance angiography performing equivalently.

Lastly, Saida et al. [76] and Cosset et al. [56] focused on identifying and classifying endoleaks. Saida et al. [76] aimed to optimize an unenhanced MRI technique using motion sensitized driven equilibrium combined with balanced turbo field echo sequences to detect and classify endoleaks in an aortic phantom, while Cosset et al. [56] investigated the feasibility of identifying and characterizing the three most common within a thoracic aorta aneurysm model using bicolour K-edge imaging with a spectral photon-counting CT system in combination with a biphasic contrast agent injection.

##### 3.3.5.5. Training

Only one study used an MCL specifically for training purposes. Meess et al. [14] simulated fenestrated EVAR in a patient-specific 3D-printed AAA phantom, allowing the clinical team to optimize their procedural strategy.

## 4. Discussion

This scoping review provides the first comprehensive overview of MCL applications in aortic research, identifying 84 eligible studies published between 1998 and 2025. These studies encompassed five major research themes: (I) (bio)mechanical characterization of the aorta, (II) hemodynamics, (III) device and product testing, (IV) diagnostics, and (V) training. A broad range of biological and synthetic aortic models were employed, including porcine, canine, ovine, bovine, and human cadaveric specimens, as well as various 3D-printed or molded synthetic phantoms, demonstrating the versatility of MCL methodology. Collectively, these models have been used to research clinically relevant questions such as aortic stiffness after stent-grafting, flow disruption in dissection, false lumen pressurization, endoleak mechanisms, device validation, and radiological performance and imaging accuracy. This shows that MCLs are not limited to engineering proof-of-concept studies but are increasingly used to explore mechanisms directly impacting endovascular outcomes and patient safety.

MCLs occupy a translational niche between computational simulations and in vivo experimentation. Unlike purely numerical models, they reproduce physiological pulsatility, tissue compliance, and boundary conditions, while avoiding the ethical and logistical challenges of animal or human studies. As such, they are highly suitable for pre-clinical evaluation of stents, other devices, and novel imaging or navigation techniques such as FORS guidance [39,45,89]. Despite these advantages, MCLs inherently isolate components of the cardiovascular system, and therefore cannot fully reproduce the global interdependence of the aorta, where local biomechanical changes propagate system-wide effects.

In addition to their mechanistic and device-focused applications, MCLs also support procedural planning and decision-making in complex aortic anatomies. Recent technological advances, including the integration of 3D-printed, patient-specific aortic phantoms with multimodal imaging platforms (e.g., 4D-flow MRI, ultrasound elastography, FORS) and increasingly sophisticated image-guided workflows, position MCLs at the forefront of precision aortic research. Patient-specific phantoms enable rehearsal of fenestrated EVAR, TEVAR, and Frozen Elephant Trunk–related steps under physiologic flow [25,63]. MCL platforms further facilitate pre-procedural simulation and trainee skill development, exemplified by the patient-specific fenestrated EVAR training model developed by Meess et al. [14] Together, these applications illustrate the translational potential of MCLs for optimizing strategy, improving operator preparedness through pre-operative simulation, and supporting complex endovascular interventions.

### Limitations

Importantly, this review also reveals a fragmented methodological landscape, with substantial variability in MCL designs, configurations, and reporting quality across studies. Essential methodological details, such as pump characteristics, compliance and resistance settings, Wind Kessel simulation, and calibration procedures, were frequently missing or inconsistently described, limiting reproducibility and cross-study comparison.

Therefore, a dedicated reporting framework is urgently needed, tailored to cardiovascular MCL research. The authors’ group is currently developing a dedicated MCL quality assessment scale to address this gap and promote reproducibility and comparability among future investigations.

Moreover, all included studies share an inherent limitation: The aorta functions as an integrated organ, so any local modification in mechanical or hemodynamic properties affects the system as a whole. This fundamental constraint contributes to discrepancies between benchtop findings and clinical-physiological observations. Future progress, therefore, requires a multidisciplinary approach that combines physiology, biomechanics, and computational modeling to bridge these gaps and advance the physiological and pathophysiological understanding of the aorta.

## 5. Conclusions

This scoping review maps current applications of mock circulation loops (MCLs) in aortic research, highlighting the value for studying (bio)mechanics, hemodynamics, devices, diagnostics, and training under controlled conditions. Despite growing use, methodological heterogeneity persists. Standard reporting is essential to enhance reproducibility and support translation from benchtop experiments to clinical practice. Progress in this field will depend on multidisciplinary approaches that couple MCL experimentation with physiological, biomechanical, and computational models to capture the inherently integrated behavior of the aorta.

## Figures and Tables

**Figure 1 biomedicines-14-00028-f001:**
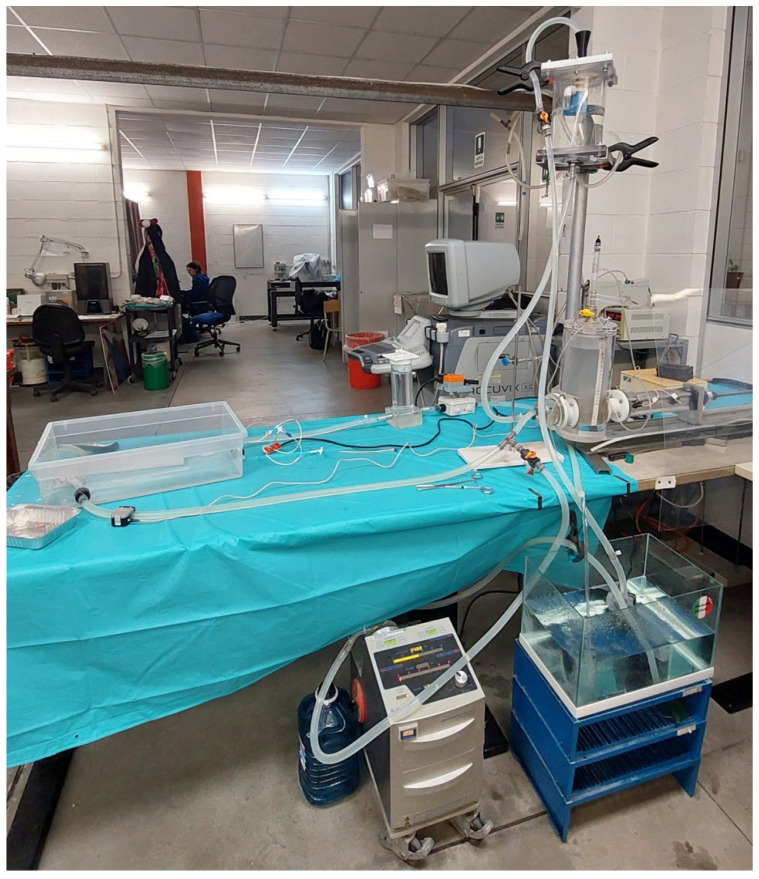
Example of a mock circulatory loop used for aortic research.

**Figure 2 biomedicines-14-00028-f002:**
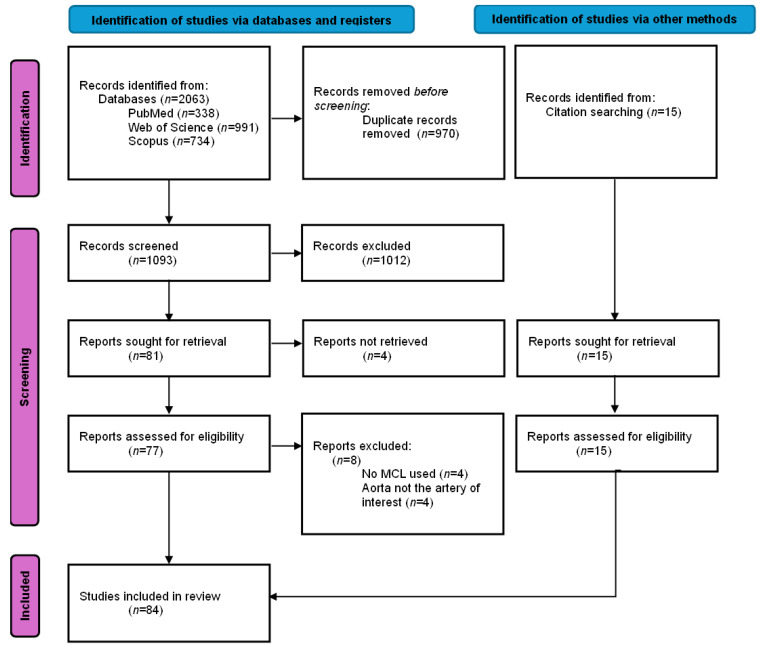
PRISMA-ScR flow diagram of the study selection process for a scoping review on mock circulatory loops used in aortic research. Note: *n* = number of articles included.

**Figure 3 biomedicines-14-00028-f003:**
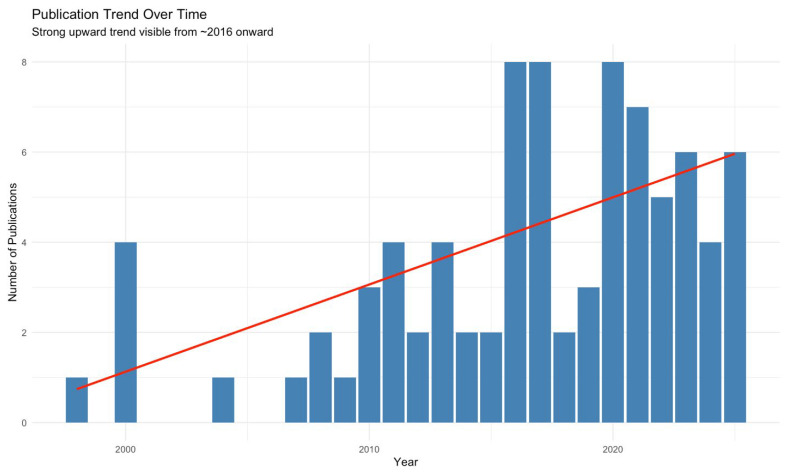
The Mann–Kendall test showing the publication trend over time of studies included in a scoping review on mock circulatory loops used in aortic research. Red line shows the parametric linear increase in number of studies published per year using MCLs.

**Table 1 biomedicines-14-00028-t001:** Categorization of studies included in a scoping review on mock circulatory loops for aortic research. Note: *n* = number of articles included.

Theme	*n*	Subcategory	*n*	Condition	*n*
(Bio)mechanical Aortic Characteristics	25	Intervention	13		
		No intervention	12	Physiologic	11
				Pathologic	1
Hemodynamics	23	Intervention	10	–	
		No intervention	13	Physiologic	2
				Pathologic	11
Products/Devices	22	Surgical	11		
		Radiological	6		
		Aortic Phantoms	5		
Diagnostics	13				
Training	1				

**Table 2 biomedicines-14-00028-t002:** Characteristics of Mock Circulatory Loops (MCLs) used in studies included in a scoping review on MCLs for aortic research.

Characteristics			*n*
Flow Pattern	Continuous		6
	Pulsatile		75
	Pulsatile and (quasi)static		2
	Unknown		1
Pump type	Centrifugal		4
	Continuous + valve		1
	Diaphragm/sac-type		8
	Gear		6
	ICS		5
	Piston		43
	Roller		3
	Unknown		14
Aorta Type	Biomaterial	Canine	1
		Human cadaveric	7
		Porcine	20
		Porcine and Ovine	1
	Synthetic		52
	Combination	Bovine and synthetic phantom	1
		Porcine and synthetic phantom	2
Working Fluid	Anti-coagulated blood		1
	Blood mimicking fluid		4
	Glycerol/glycerin		33
	Gelofusine		1
	Starch solution		1
	Saline buffered		14
	Water based		22
	Unknown		8

## Data Availability

The original contributions presented in this study are included in the article/Appendix A. Further inquiries can be directed to the corresponding author.

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
