# Peer review of "The Current State of Mock Circulatory Loop Applications in Aortic and Cardiovascular Research: A Scoping Review"

_biomedicines, 2025, doi:10.3390/biomedicines14010028_

Round 1

Reviewer 1 Report

Comments and Suggestions for Authors

This study is commendable for its reproducible methodology and clear scientific rigor. The authors employed a well-structured scoping review approach that is both innovative and thoughtfully aligned with the study's objectives. The literature is up-to-date and appropriate. The results are robust and convincingly presented, with strong relevance to the broader field. The discussion is insightful, connecting the findings to existing literature while also highlighting the study's unique contributions. The clarity of the data presentation reflects careful planning and attention to detail. Overall, this paper presents high-impact results with excellent data  interpretation, making it a valuable addition to current scientific discourse and future research directions.

here are my additional comments.

The main question addressed by the research is to comprehensively chart the existing landscape of MCLs applied in aortic simulation and pinpoint key areas of their usage. This topic is original and relevant to the field, it does address an important knowledge gap in the field. It integrates all MCL related research and emphasizes the importance of standardized reporting frameworks to enhance reproducibility and speed up the translation to personalized patient planning in clinical practices. The methodology employed is of standard and the method description enables reproducible endeavors by others. The conclusions are consistent with the evidence and arguments presented and do they address the main question posed in this field. The references are appropriate and up to date. The tables and figures are nicely constructed and structured. Thank you.

Author Response

Reviewer 1:

This study is commendable for its reproducible methodology and clear scientific rigor. The authors employed a well-structured scoping review approach that is both innovative and thoughtfully aligned with the study's objectives. The literature is up-to-date and appropriate. The results are robust and convincingly presented, with strong relevance to the broader field. The discussion is insightful, connecting the findings to existing literature while also highlighting the study's unique contributions. The clarity of the data presentation reflects careful planning and attention to detail. Overall, this paper presents high-impact results with excellent data  interpretation, making it a valuable addition to current scientific discourse and future research directions.

here are my additional comments.

The main question addressed by the research is to comprehensively chart the existing landscape of MCLs applied in aortic simulation and pinpoint key areas of their usage. This topic is original and relevant to the field, it does address an important knowledge gap in the field. It integrates all MCL related research and emphasizes the importance of standardized reporting frameworks to enhance reproducibility and speed up the translation to personalized patient planning in clinical practices. The methodology employed is of standard and the method description enables reproducible endeavors by others. The conclusions are consistent with the evidence and arguments presented and do they address the main question posed in this field. The references are appropriate and up to date. The tables and figures are nicely constructed and structured. Thank you.

We thank the reviewer for the thorough and positive evaluation of our work. We appreciate the recognition of the study’s methodology, clarity, and relevance to the field. We are grateful for the reviewer’s supportive comments regarding the originality, robustness, and scientific contribution of this scoping review.

Changes to the manuscript:

None

Reviewer 2 Report

Comments and Suggestions for Authors

This review article is about how mock circulatory loops (MCLs) are being used in aortic research. These systems provide a flexible way to study how the aorta behaves and to test different endovascular treatments in a controlled setting. The review also highlights the need for standard guidelines so that results are consistent and can more easily be applied to patient-specific planning.

Research design is good. Language of the paper is fine.

Reconsider citation with [  ] and not with (  ).

If possible, discussion section should be extended because, it is a review article.

At certain places, proper citation is missing. For example; On line 447, “Lastly, Saida et al. and Cosset et al.”

Rest of the document is fine. This review article may be accepted after incorporating these minor changes.

Author Response

Reviewer 2:

This review article is about how mock circulatory loops (MCLs) are being used in aortic research. These systems provide a flexible way to study how the aorta behaves and to test different endovascular treatments in a controlled setting. The review also highlights the need for standard guidelines so that results are consistent and can more easily be applied to patient-specific planning.

Research design is good. Language of the paper is fine.

We thank the reviewer for the positive assessment of our work and appreciate the supportive remarks regarding the research design and language quality. We are pleased that the relevance and value of synthesising current MCL applications and highlighting the need for standardised reporting were recognised.

  1. Reconsider citation with [  ] and not with (  ).

We thank the reviewer for this suggestion. The citation style has been revised throughout the manuscript, and all in-text references now follow the bracketed format as requested.

Changes to the manuscript:

Changed all references (.) to [.]

  1. If possible, discussion section should be extended because, it is a review article.

We thank the reviewer for this helpful suggestion. The Discussion section has now been expanded to provide greater depth, including additional context on translational relevance, methodological heterogeneity, and future directions for standardisation and clinical integration.

Changes to the manuscript:

Expanded the discussion section.

  1. At certain places, proper citation is missing. For example; On line 447, “Lastly, Saida et al. and Cosset et al.”

We thank the reviewer for pointing this out. The missing citations have now been added, including the full references for Saida et al. and Cosset et al. at the indicated location. We also re-checked the entire manuscript to ensure that all similar instances are properly cited.

Changes to the manuscript:

Re-checked complete manuscript to ensure proper citing.

Rest of the document is fine. This review article may be accepted after incorporating these minor changes.

Reviewer 3 Report

Comments and Suggestions for Authors

This scoping review presents a timely and comprehensive work that systematically describes the application of physical mock circulatory loops (MCLs) in aortic research. The relevance of the topic is undeniable, given the growing role of MCLs as a bridge between computational modeling and in vivo studies.

The authors conducted a thorough systematic literature search and selection, including 84 studies, which allows for a representative picture of the field. The use of pre-defined thematic categories ((bio)mechanics, hemodynamics, device testing, diagnostics, training) contributes to a clear structuring of the results.

The work addresses an existing gap in the literature by providing the first comprehensive overview of the use of MCLs for aortic research. The growth in publication activity after 2016 demonstrates the dynamic development of this area.

The review analyzes the key components of MCLs (types of pumps, aortic models, working fluids), which is highly valuable for researchers planning to develop or use such systems.

In the "Discussion" section, the key problem of the field is rightly identified—methodological heterogeneity and the lack of a systematic approach in conducting research, which leads to inconsistencies in the results obtained.

Comment:
Incompleteness of the conclusion: The authors fail to note a fundamental shortcoming common to all the studies reviewed. This shortcoming lies in the fact that the aorta, being an integral organ, changes as a whole with any local alteration in its properties (biomechanics or hemodynamics). This limitation of all the presented studies largely accounts for the discrepancies that arise when comparing clinical-physiological observations with methods of physical and numerical modeling.

Therefore, the conclusion should postulate the necessity of a multidisciplinary approach to fill the gaps in the physiology and pathophysiology of the aorta.

The presented scoping review is a high-quality and significant work that will be useful to a wide range of specialists—from surgeons and cardiologists to biomedical engineers and researchers. It summarizes the current state of the field, and clearly defines directions for future development. The article deserves high praise and publication.

Author Response

Reviewer 3:

This scoping review presents a timely and comprehensive work that systematically describes the application of physical mock circulatory loops (MCLs) in aortic research. The relevance of the topic is undeniable, given the growing role of MCLs as a bridge between computational modeling and in vivo studies.

The authors conducted a thorough systematic literature search and selection, including 84 studies, which allows for a representative picture of the field. The use of pre-defined thematic categories ((bio)mechanics, hemodynamics, device testing, diagnostics, training) contributes to a clear structuring of the results.

The work addresses an existing gap in the literature by providing the first comprehensive overview of the use of MCLs for aortic research. The growth in publication activity after 2016 demonstrates the dynamic development of this area.

The review analyzes the key components of MCLs (types of pumps, aortic models, working fluids), which is highly valuable for researchers planning to develop or use such systems.

In the "Discussion" section, the key problem of the field is rightly identified—methodological heterogeneity and the lack of a systematic approach in conducting research, which leads to inconsistencies in the results obtained.

We thank the reviewer for the positive and thoughtful evaluation of our work. We appreciate the recognition of the comprehensive search strategy, thematic structuring, and the relevance of mapping current applications of MCLs in aortic research. We are pleased that the reviewer highlights the value of our analysis of MCL components and agrees with the identification of methodological heterogeneity as a key challenge in the field.

Comment:
1. Incompleteness of the conclusion: 

The authors fail to note a fundamental shortcoming common to all the studies reviewed. This shortcoming lies in the fact that the aorta, being an integral organ, changes as a whole with any local alteration in its properties (biomechanics or hemodynamics). This limitation of all the presented studies largely accounts for the discrepancies that arise when comparing clinical-physiological observations with methods of physical and numerical modeling. Therefore, the conclusion should postulate the necessity of a multidisciplinary approach to fill the gaps in the physiology and pathophysiology of the aorta.

We thank the reviewer for highlighting this important conceptual limitation. In response, we have strengthened the manuscript by integrating this point into both the Discussion and Limitations sections. We now explicitly acknowledge that MCLs, by design, isolate components of the cardiovascular system and therefore cannot fully replicate the global, integrated behaviour of the aorta. We further emphasise that this intrinsic constraint contributes to discrepancies between benchtop findings and clinical physiology.
Additionally, we highlight in the Conclusion that future progress will require multidisciplinary approaches combining physiological, biomechanical, and computational perspectives. Together, these revisions address the reviewer’s valuable suggestion and improve the clarity and completeness of the manuscript.

Changes to the manuscript:

L 465: “Despite these advantages, MCLs inherently isolate components of the cardiovascular system, and therefore cannot fully reproduce the global interdependence of the aorta, where local biomechanical changes propagate system-wide effects”

L 491: “Moreover, all included studies share an inherent limitation: the aorta functions as an integrated organ, and any local modification in mechanical or hemodynamic properties affects the system as a whole. This fundamental constraint contributes to discrepancies between benchtop findings and clinical-physiological observations. Future progress therefore requires a multidisciplinary approach that combines physiology, biomechanics, and computational modeling to bridge these gaps and advance the physiological and pathophysiological understanding of the aorta.”

L 504: “Progress in this field will depend on multidisciplinary approaches that couple MCL experimentation with physiological, biomechanical, and computational models to capture the inherently integrated behaviour of the aorta.”

The presented scoping review is a high-quality and significant work that will be useful to a wide range of specialists—from surgeons and cardiologists to biomedical engineers and researchers. It summarizes the current state of the field, and clearly defines directions for future development. The article deserves high praise and publication.

Reviewer 4 Report

Comments and Suggestions for Authors

MAJOR POINTS

– Summarizes the transition toward integrated imaging ecosystems, AI-driven workflow augmentation, and multimodal fusion (echo + CT + MRI + 3D modeling).

-The Discussion outlines the role of MCLs in translational research but does not elaborate on their impact on: procedural planning (fenestrated EVAR, TEVAR, FET), decision-making in complex aortic anatomies, pre-procedural simulation for trainees and interventionalists.

MINOR POINTS

- typos: Line 55: “benchtop systems that simulate…” (sentence structure could be improved), Line 459–462: spacing before parentheses is inconsistent.

Author Response

Reviewer 4:

We thank the reviewer for taking their time to review our manuscript.

MAJOR POINTS

  1. Summarizes the transition toward integrated imaging ecosystems, AI-driven workflow augmentation, and multimodal fusion (echo + CT + MRI + 3D modeling).

We thank the reviewer for this observation. The broader transition toward integrated imaging ecosystems, multimodal fusion, and increasingly sophisticated workflow augmentation is indeed an important development in cardiovascular research. We have now strengthened the Discussion to reflect this trend more explicitly, highlighting recent advances that combine patient-specific phantoms with multimodal imaging platforms and enhanced image-guided workflows.

Changes to the manuscript:

L 469: “Recent technological advances, including the  integration of 3D-printed, patient-specific aortic phantoms with multimodal imaging platforms (e.g. 4D-flow MRI, ultrasound elastography, FORS) and increasingly sophisticated image-guided workflows, position MCLs at the forefront of precision aortic research.”

  1. The Discussion outlines the role of MCLs in translational research but does not elaborate on their impact on: procedural planning (fenestrated EVAR, TEVAR, FET), decision-making in complex aortic anatomies, pre-procedural simulation for trainees and interventionalists.

We thank the reviewer for this helpful comment. We agree that the role of MCLs in procedural planning, decision-making in complex aortic anatomies, and pre-procedural simulation is an important aspect of their translational value. These applications are now explicitly addressed in the Discussion.

Changes to the anuscript:

L 473: “Patient-specific phantoms enable rehearsal of fenestrated EVAR, TEVAR, and Frozen Elephant Trunk–related steps under physiologic flow. [25,63] MCL platforms further facilitate pre-procedural simulation and trainee skill development, exemplified by the patient-specific fenestrated EVAR training model developed by Meess et al. [14] Together, these applications illustrate the translational potential of MCLs for optimizing strategy, improving operator preparedness through pre-operative simulation, and supporting complex endovascular interventions.”

MINOR POINTS

  1. - typos: Line 55: “benchtop systems that simulate…” (sentence structure could be improved),
  2.  Line 459–462: spacing before parentheses is inconsistent.

 We thank the reviewer for these observations. The sentence structure at line 55 has been revised for clarity, and the spacing inconsistencies noted at lines 459–462 have been corrected.

L 20: “Mock circulatory loops (MCLs) are benchtop experimental platforms that reproduce key features of the human cardiovascular system, providing a safe, controlled, and reproducible environment for haemodynamic investigation.”

Round 2

Reviewer 4 Report

Comments and Suggestions for Authors

At Line 472 : "(e.g. 4D-flow MRI, ultrasound elastography, FORS) and increasingly sophisticated image-guided workflows" add appropriate citation : Holographic mixed reality for planning transcatheter aortic valve replacement. Int J Cardiol. 2024 Oct 1;412:132330. doi: 10.1016/j.ijcard.2024.132330.

Author Response

At Line 472 : "(e.g. 4D-flow MRI, ultrasound elastography, FORS) and increasingly sophisticated image-guided workflows" add appropriate citation : Holographic mixed reality for planning transcatheter aortic valve replacement. Int J Cardiol. 2024 Oct 1;412:132330. doi: 10.1016/j.ijcard.2024.132330.
We thank the reviewer for this helpful suggestion. We have now added the recommended citation to support the statement regarding advanced image-guided workflows.

Changes to the manuscript:

L471: Recent technological advances, including the integration of 3D-printed, patient-specific aortic phantoms with multimodal imaging platforms (e.g. 4D-flow MRI, ultrasound elastography, FORS) and increasingly sophisticated image-guided workflows, position MCLs at the forefront of precision aortic research. [39,45,89